# Cohort Profile: Childhood morbidity and potential non-specific effects of the childhood vaccination programmes in the Nordic countries (NONSEnse): register-based cohort of children born 1990–2017/2018

Lise Gehrt [ID],[1,2] Ida Laake,[3] Hélène Englund [ID],[4] Heta Nieminen,[5] Berit Feiring,[3] Mika Lahdenkari,[6] Arto A Palmu,[5] Lill Trogstad,[3] Christine Stabell Benn,[1,2] Signe Sørup [ID] [1,7]

For numbered affiliations see end of article.

**Correspondence to**
MSc Hélène Englund;
helene.englund@
folkhalsomyndigheten.se

## ABSTRACT

**Purpose** The aim of the NONSEnse project is to investigate the non-specific effects of vaccines and immunisation programmes on the overall health of children by using information from the extensive nationwide registers on health and sociodemographic factors in Denmark, Finland, Norway and Sweden.

**Participants** The cohort covers 9 072 420 children aged 0–17 years, born 1990–2017/2018 and living in Denmark, Finland, Norway or Sweden. All countries use a unique identification number for its permanent residents, which makes it possible to link individual-level information from different registers.

**Findings to date** Data collection and harmonisation according to a common data model was completed in March 2022. As a prerequisite for comparing the effects of childhood vaccinations on the overall health of children across the Nordic countries, we have identified indicators measuring similar levels of infectious disease morbidity across these settings. So far, studies pertaining to non-specific effects of vaccines are limited to investigations that could be undertaken using aggregated data sets that were available before the NONSEnse cohort with individual-level information was completely set up.

**Future plans** We are currently performing several studies of the effects on non-targeted infectious disease morbidity across the countries following vaccination against measles, mumps, rubella, diphtheria, tetanus, pertussis, human papillomavirus, rotavirus and influenza. Multiple studies are planned within the next years using different study designs to facilitate triangulation of results and enhance causal inference.

**Registration** No clinical trials will be conducted within the NONSEnse project.

## STRENGTHS AND LIMITATIONS OF THIS STUDY

⇒ Complete population cohort minimises selection bias.
⇒ Real-world data which have been collected, collated and quality checked.
⇒ A common data model enables uniform data analysis across countries.
⇒ Lacking information on some potential confounding factors.
⇒ The data harmonisation process may entail loss of details of country-specific data.

protection, may have so-called non-specific effects affecting susceptibility towards other diseases than the vaccine-targeted infections.[1 2] Most previous studies on non-specific effects stem from low-income countries with a high infectious disease burden and have had overall childhood mortality as the outcome. The non-specific effects are found to vary depending on the type of vaccine being administered. Live vaccines have been associated with beneficial non-specific effects.[1 2] Non-live vaccines, although protecting against the vaccine-targeted infections, may possibly increase susceptibility to other infections.[1 2] The effects are most pronounced for the most recently administered vaccine.[2]

Studies of non-specific effects from high-income countries have primarily focused on infectious disease morbidity[3] and atopic diseases.[4–8] Most of these studies are observational because it would often be unethical to randomise children to refrain from or delay recommended childhood vaccinations. Therefore, concerns about different types

## INTRODUCTION

An accumulating number of epidemiological and immunological studies have found that vaccines, in addition to the disease-specific

of bias in different settings and observational designs have been raised.[9–11] Triangulation has been proposed as a method to strengthen causal inference in epidemiology by integrating results from several epidemiological designs and between different populations with different bias structures while using the same analysis plan across settings to enhance comparability of results.[12 13]

The 'NONSEnse' project is a NordForsk-funded collaboration between research groups in the four Northern European countries Denmark, Finland, Norway and Sweden (henceforth referred to as the Nordic countries). The main aim of NONSEnse is to evaluate if childhood vaccinations influence other health outcomes than those targeted by the vaccine in the Nordic countries. The main hypothesis underlying this evaluation is that having a live vaccine as the most recent vaccine is associated with beneficial non-specific effects and thus a lower morbidity in the following time period, compared with having a non-live vaccine as the most recent vaccine. The individual studies will be undertaken using the same methodology and statistical coding across countries. Furthermore, we will examine the same research question in multiple studies using different analytical approaches to facilitate triangulation of the results. The main associations we will examine are associations between childhood vaccinations and (1) infectious disease hospitalisations, (2) antibiotic use and (3) atopic diseases (asthma, atopic dermatitis, allergic rhinoconjunctivitis).

The first step has been to examine and compare infectious disease and atopic morbidity among children in the respective countries over time and by age and sex, to inform choice of design and outcome definitions in the subsequent studies of non-specific effects of vaccines.

The aim of the present cohort profile is to describe the content and quality of the data included in the registry-based NONSEnse cohort and present characteristics of the cohort, thereby demonstrating the research potential of the NONSEnse cohort. The insights presented can be used to guide future epidemiological research projects using registry data from the Nordic countries.

## COHORT DESCRIPTION
### Setting
The Nordic countries have many similarities including the welfare state model with universal tax-funded healthcare and a high level of social security. A detailed description of the Nordic healthcare systems and basic demographics has been published elsewhere.[14]

### National immunisation programmes
Childhood vaccinations within the national immunisation programmes (NIP) are voluntary and administered free of charge in all four countries. In Denmark, all childhood vaccines are administered by family practitioners.[15] In Finland, Norway and Sweden, vaccines scheduled before school age are administered at well-baby clinics by nurses; during school age, the vaccines are administered by school nurses.[16–18] In 2018, children were offered vaccinations against 10 diseases in Denmark,[15] up to 13 diseases in Finland,[16] 12 diseases in Norway[18] and 10 diseases in Sweden.[17] Children in specific risk groups are offered vaccines against additional diseases according to national guidelines.[16–19] An overview of recommended childhood vaccinations in the four countries in 2018 is presented in table 1 and historical changes are illustrated in online supplemental appendix 1.

### Nordic nationwide register data: a goldmine for epidemiological studies
All individuals residing in the Nordic countries are assigned a unique personal identification (ID) code. All four countries have extensive national registers on health, demographic factors and socioeconomic factors collected for administrative purposes and linked to the individual using the personal ID.[14 20] The register information is collected automatically, which minimises systematic reporting bias, for example, recall bias. The use of national registers limits selection bias as the entire population is included. All information in the registers is dated, which ensures that exposures and outcomes can be temporally linked and facilitates investigation of the cumulative and combined effects of multiple interventions on childhood health. Thus, the structure of the Nordic registers presents a unique opportunity to investigate the real-life effects of childhood vaccinations while incorporating multiple potential confounding factors.

### Study population
We used national population registers to identify all children aged 0–17 years, who were born or became permanent residents after migrating to one of the Nordic countries at some point from 1990 until and including 2018 in Denmark and Norway, and 2017 in Finland and Sweden[21–24] (figure 1). End of follow-up in each country reflects when the data application process was final. The individual registries included in this cohort were established in the respective countries at different time points. We have included the birth cohorts from 1990 in all countries to ensure that we have full information on follow-up from birth also for the children who will be included at older ages for, example, the studies of human papillomavirus (HPV) vaccination given to teenagers. The population data obtained in Finland had incomplete information on migration history before 2014 and thus we were unable to assess the date of entering the country for children born abroad. As a result, we limited the Finnish study population to children born in the country to ensure that they were present in the country from the beginning of follow-up. After exclusions, which were primarily due to uncertain information about residency, a total of 9 072 420 children were included across the countries (figure 1). Children were followed from date of birth or date of immigration until the date of first emigration, 18-year birthday, death or last date with available information, whichever came first.

**Table 1** Vaccines recommended to children in Denmark, Finland, Norway and Sweden in 2018

| Disease (vaccine) | Denmark | Finland | Norway | Sweden |
|---|---|---|---|---|
| Tuberculosis (BCG) | Not within programme | Before 7 years of age, risk groups only* | 6 weeks of age, risk groups only* | After 6 months of age, risk groups only*† |
| Hepatitis A | Not within programme | From 1 year of age, risk groups only‡ | Not within programme | Not within programme† |
| Hepatitis B | From birth, risk groups only§ | From birth, risk groups only§¶ | 3 doses: 3, 5, 12 months of age | Not within programme but recommended to all children. 3 doses: 3, 5, 12 months of age** |
| Rotavirus | Not within programme | 3 doses: 2, 3, 5 months of age | 2 doses: 6 weeks, 3 months of age | 2 or 3 doses: 6 weeks, 3 and 5 months of age†,†† |
| Diphtheria, tetanus and pertussis (DTaP) | 4 doses: 3, 5, 12 months, booster at 5 years of age | 5 doses: 3, 5, 12 months of age, booster at 4 and 14 years of age | 5 doses: 3, 5, 12 months of age, booster in 2nd and 10th school years | 5 doses: 3, 5, 12 months of age, booster at 5 years of age and in 8th or 9th school year |
| Polio (IPV) | 4 doses: 3, 5, 12 months, booster at 5 years of age | 4 doses: 3, 5, 12 months of age, booster at 4 years of age | 5 doses: 3, 5, 12 months of age, booster in 2nd and 10th school years | 4 doses: 3, 5 and 12 months of age, booster at 5 years of age |
| *Haemophilus influenzae* type B | 3 doses: 3, 5, 12 months of age | 3 doses: 3, 5, 12 months of age | 3 doses: 3, 5, 12 months | 3 doses: 3, 5 and 12 months of age |
| Pneumococcal disease (PCV) | 13-valent; 3 doses: 3, 5, 12 months of age | 10-valent; 3 doses: 3, 5, 12 months of age | 13-valent; 3 doses: 3, 5, 12 months of age | 10 or 13-valent; 3 doses: 3, 5 and 12 months of age |
| Influenza (live or non-live influenza vaccine) | From 6 months of age, risk groups only‡‡ | Yearly, from 6 months to 6 years of age and for risk groups after 6 years of age‡‡ | From 6 months of age, risk groups only, through the influenza immunisation programme‡‡ | Yearly, from 6 months of age, risk groups only†‡‡ |
| Measles, mumps and rubella | 2 doses: 15 months of age and 4 years of age | 2 doses: 12 months of age, 6 years of age | 2 doses: 15 months of age, and 6th school year | 2 doses: 18 months of age and 1st or 2nd school year |
| Varicella | Not recommended | 1.5–11 years of age | Not recommended | Not recommended |
| Pneumococcal disease (PPV) | Not within programme | Before 5 years of age, after PCV, risk groups only§§ | Not within programme, but recommended from 2 years of age, to specified risk groups§§ | Not within programme, but recommended from 2 years of age, to specified risk groups†§§ |
| Tickborne encephalitis | Not within programme | From 3 years of age, risk groups only¶¶ | Not within programme | Not within programme |
| Human papillomavirus | 2 doses: 12 years of age, girls only | 2 doses: 6th school year, girls only | 2 doses in 7th school year | 2 doses in 5th or 6th school year, girls only |

The vaccines are included in the childhood immunisation programmes and registered in the vaccination registers, unless otherwise specified. Information obtained from: Danish Health Authority,[15] Finnish Institute for Health and Welfare,[16] Norwegian Institute of Public Health[18] and Public Health Agency of Sweden.[17]
*Children with a parent from a country with a high incidence of tuberculosis.
†Not included in the vaccination registry.
‡Children of intravenous drug users.
§(1) Children of mothers or another member of the household who are hepatitis B positive, or (2) attend day care with a child who has hepatitis B.[19 45]
¶(1) Children of parents from countries with high incidence of hepatitis B, or (2) children of mothers with hepatitis C infection.[45]
**Only offered to children in the risk group before 2016, not included in the vaccination registry before 2016.[46]
††Rotavirus vaccine was offered by some Swedish regions as part of regional vaccination schemes.
‡‡Children with increased risk of severe influenza illness or members of households with high-risk individuals.[17 47–49]
§§Children with increased risk of severe pneumococcal disease, for example, children with chronic diseases.[17 50 51]
¶¶Children of families with a permanent home or holiday house in areas within Finland with high tick prevalence.[52]
BCG, Bacillus Calmette-Guérin vaccine; DTaP, diphtheria, tetanus and acellular pertussis vaccine; IPV, inactivated polio vaccine; PCV, pneumococcal conjugated vaccine; PPV, pneumococcal polysaccharide vaccine.

## Source and content of data

Using the personal ID, we linked information from the nationwide registers and obtained individual-level information on gestation and birth, hospital contacts, redeemed prescriptions and receipt of childhood vaccines. Furthermore, each child was linked to their parents through the population registers in order to extract information on household income, family composition and highest attained parental education (figure 2). The included data reflect necessary information to identify the vaccination status of the child, relevant outcomes, potential confounding factors and information to be included as negative control outcomes.

Information on administered vaccines including type of vaccine and date of vaccination was obtained from the Danish Vaccination Register in Denmark,[25] the Finnish

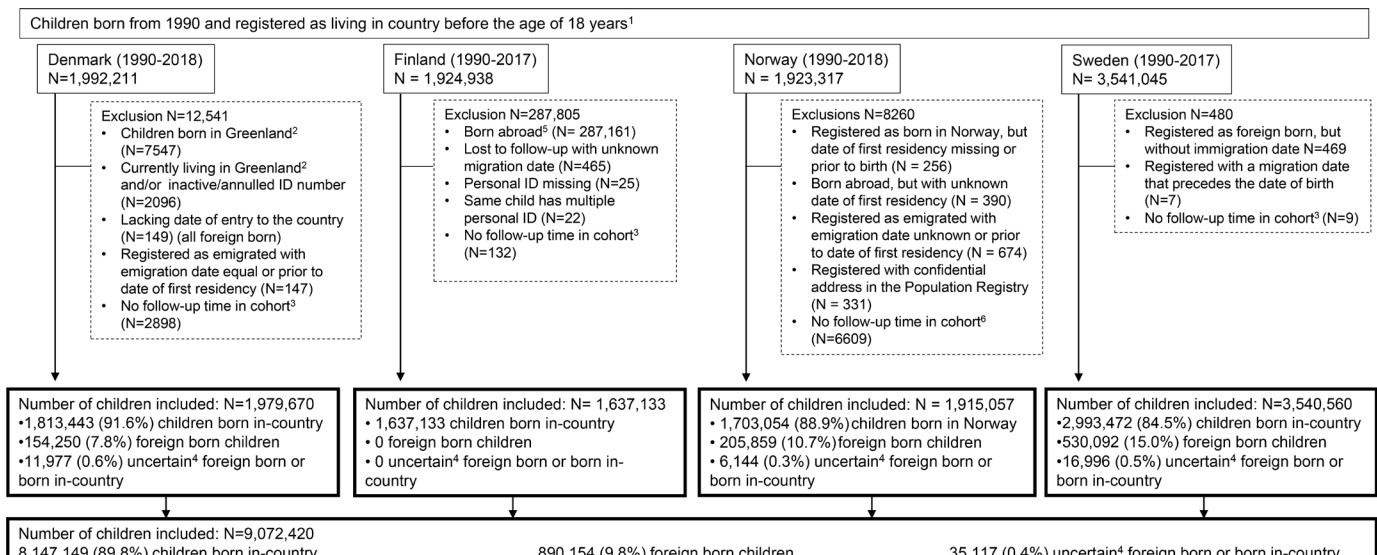

**Figure 1** Flowchart of study populations in Denmark, Finland, Norway and Sweden. [1]In Finland, data from the population register including information on deaths and migrations were obtained on 8 February 2014. Thus, only children who were still alive and living in the country from this date were included. [2]Children born or residing in Greenland are registered as living in Denmark. However, the Greenlandic hospitals and pharmacies do not report to the patient register or prescription register. [3]Children who die or migrate on the same date as they enter the cohort. [4]Children registered as born in the country but with an immigration date registered without a preceding emigration date: in these cases, it is not clear if the child is born in-country or has immigrated to the country. [5]Most immigration dates were not known, thus all children born abroad were excluded. [6]Date of birth was assigned as a random integer within the month of birth, thus children with date of death or migration within the month of birth is regarded as having no follow-up time in the cohort.

Vaccination Register in Finland,[26] the Norwegian Immunisation Registry in Norway[27] and the National Vaccination Register in Sweden.[28] Registration of vaccinations within the NIP is mandatory in all Nordic countries (table 1).

The Danish Vaccination Register includes information from the Danish National Health Insurance System that collects information on all vaccinations within the NIP.[29] Since 2015, it has also been mandatory to report on vaccines given outside the NIP.[30] In Denmark, vaccine information is linked to the individual using the personal ID; however, before 1997 the information was registered on the ID of the parents only.[29] Thus, in Denmark, only information on vaccines administered from 1997 and later was included. In Finland, the register includes all vaccines given in public healthcare since 2009, and after 2016 also private healthcare is obligated to register vaccinations.[26] In Norway, the immunisation registry holds information since 1995 on all administered vaccines that are part of the NIP.[18] Since 2011, notification to the immunisation registry is also mandatory for vaccines given outside the NIP.[27] The Swedish National Vaccination Register has information about vaccinations given since 2013, but only those included in the NIP.[28]

Information on hospital contacts was obtained from the Danish National Patient Register, Finnish Care Register for Health Care, Norwegian National Patient Register and the Swedish Patient Registry.[14 20] The registries reached national coverage and recorded individual-level data since 1978 in Denmark, 1994 in Finland, 2008 in Norway and 1997 in Sweden. Since 1997, diagnoses have been coded according to the International Classification of Diseases version 10 in all four countries.[31]

The Danish, Norwegian and Swedish prescription registers hold information on all redeemed prescriptions, classified using the Anatomical Therapeutic Chemical (ATC) classification system since 1995, 2004 and mid-2005, respectively.[32] The Finnish Benefits Registry holds information only for reimbursable redeemed prescriptions.[32–34] In addition, the Finnish Prescription Center started gradually in 2010 and collects all redeemed prescriptions irrespective of reimbursement. By 2017, practically all prescriptions were included in the Finnish Prescription Center.[35] We combined the information from the Finnish Prescription Center and the Finnish Benefits Registry to obtain the most complete information on redeemed prescriptions (see online supplemental appendix 2 for details on source of data).

Information on socioeconomic factors and birth characteristics was available from the beginning of the study period (1990) in all countries.

### The common data model: harmonised country-specific data sets
The country-specific data from the national registers may differ both across countries and within countries over time due to differences in coding practices, administration and country-specific legislation on health and social aspects.[20] We developed a common data model to harmonise all information we obtained into similar data sets using the same variable names and same categories in all

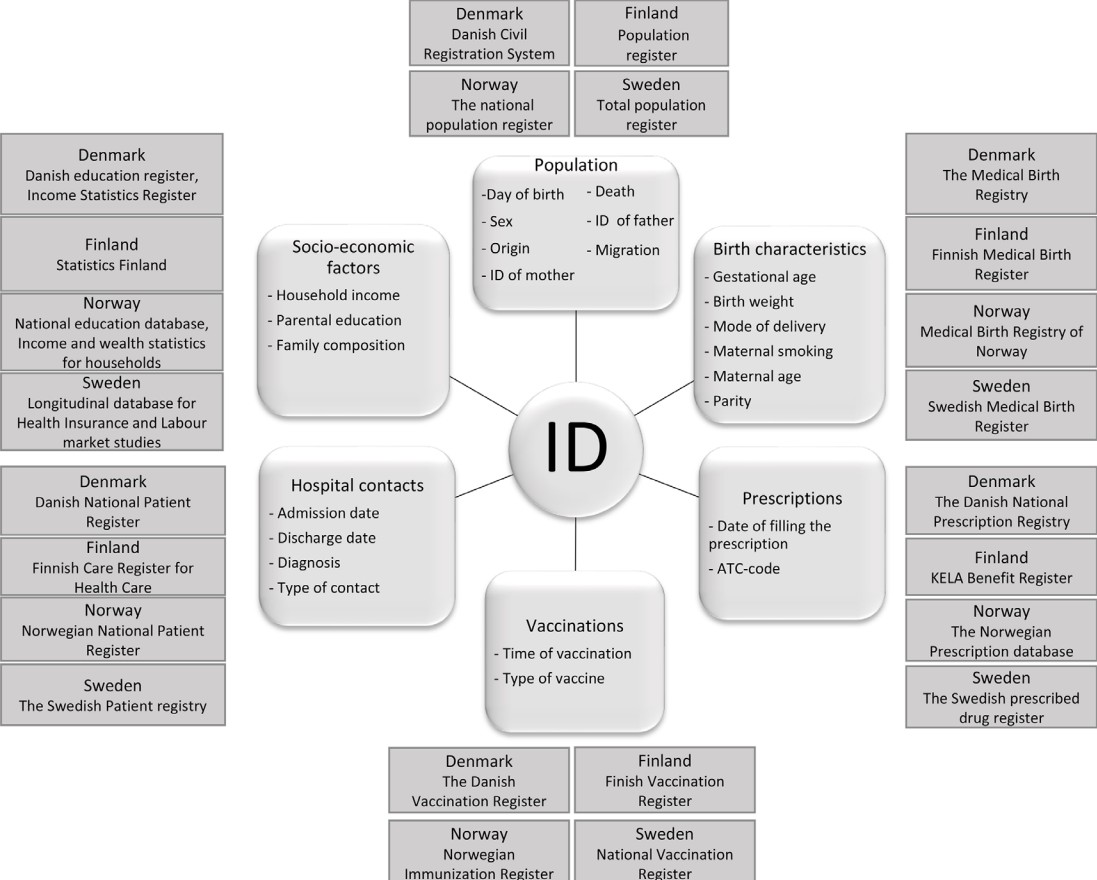

**Figure 2** Nordic register information linked to the individual using a unique personal identification (ID) code. ATC, Anatomical Therapeutic Chemical classification system.

four countries (figure 3). The data harmonisation focused on identifying outliers and country-specific traits that could hinder cross-country comparability. Information on source of data and data preparation for each of the variables can be found in online supplemental appendix 2 'NONSense Common Data Model'.

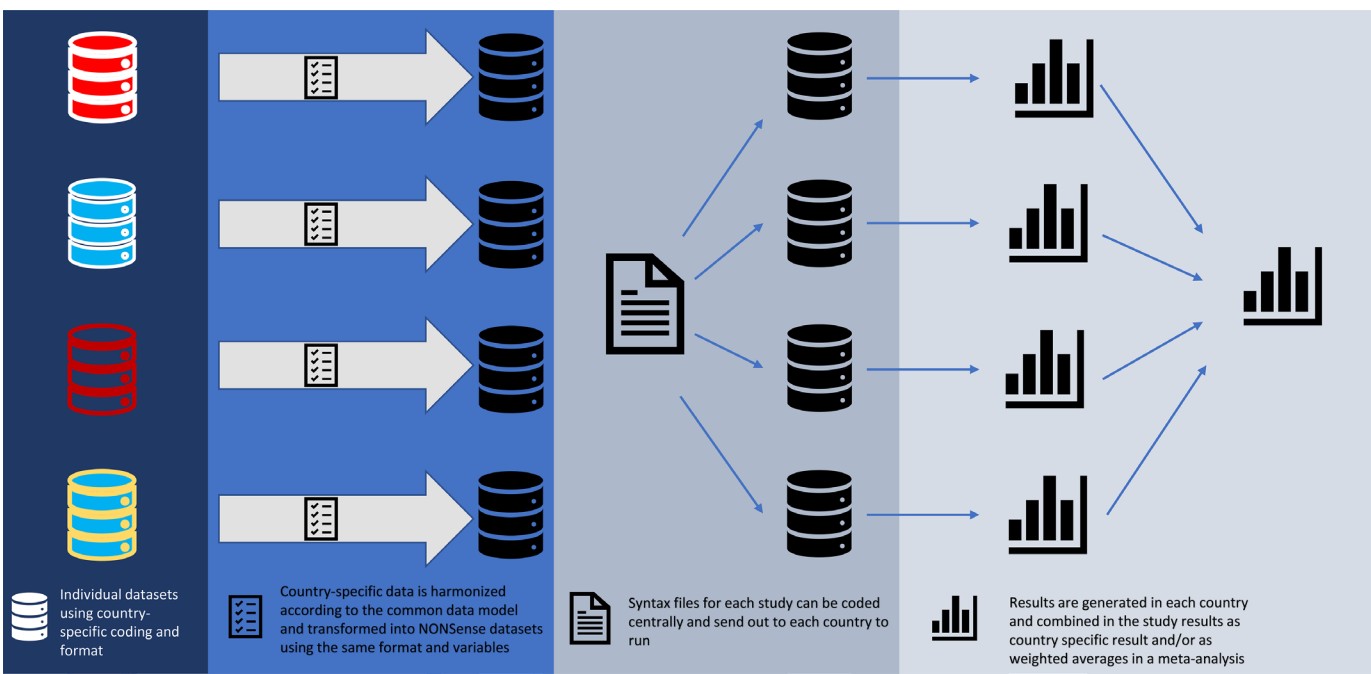

**Figure 3** Transforming country-specific data sets into NONSEnse data sets using a common data model.

**Table 2** Study population—identification and follow-up

| | Denmark | Finland* | Norway | Sweden |
|---|---|---|---|---|
| Study population (n) | 1 979 670 | 1 637 133 | 1 915 057 | 3 540 560 |
| Years of follow-up† per child, median (p25–p75) | 13.1 (5.9–18.0) | 14.2 (7.2–18.0) | 12.6 (5.7–18.0) | 10.8 (4.2–18.0) |
| Year of birth | 1990–2018 | 1990–2017 | 1990–2018 | 1990–2017 |
| Sex, n (%) | | | | |
| Male | 1 014 745 (51.3) | 836 828 (51.1) | 985 568 (51.5) | 1 827 619 (51.6) |
| Female | 964 925 (48.7) | 800 305 (48.9) | 929 489 (48.5) | 1 712 941 (48.4) |
| Reason for entering the cohort, n (%) | | | | |
| Birth | 1 813 443 (91.6) | 1 637 133 (100.0) | 1 703 054 (88.9) | 2 993 472 (84.5) |
| Immigration | 166 227 (8.4) | 0 (0.0) | 212 003 (11.1) | 547 088 (15.5) |
| Reason for leaving the cohort, n (%) | | | | |
| Death | 8532 (0.4) | 761 (0.0) | 5422 (0.3) | 5614 (0.2) |
| Emigration | 122 916 (6.2) | 11 789 (0.7) | 95 406 (5.0) | 154 878 (4.4) |
| Other‡ | 1917 (0.1) | 0 (0.0) | 0 (0.0) | 0 (0.0) |
| 18th birthday | 704 518 (35.6) | 608 644 (37.2) | 703 164 (36.7) | 1 280 027 (36.2) |
| End of follow-up§ | 1 141 787 (57.7) | 1 015 939 (62.1) | 1 111 065 (58.0) | 2 100 041 (59.3) |
| Linked with mother in registers, n (%) | 1 961 595 (99.1) | 1 634 120 (99.8) | 1 894 916 (98.9) | 3 352 706 (94.7) |
| Linked with father in registers, n (%) | 1 920 008 (97.0) | 1 601 138 (97.8) | 1 838 444 (96.0) | 3 248 108 (91.7) |
| Maternal age at birth of child, median (p25–p75) | 29 (26–33) | 29 (26–33) | 29 (25–33) | 29 (26–33) |
| Missing information on maternal age, n (%) | 18 075 (0.9) | 10 099 (0.6) | 20 141 (1.1) | 187 854 (5.3) |
| Maternal origin, n (%) | | | | |
| Born in-country | 1 582 885 (80.0) | 1 520 159 (92.9) | 1 432 179 (74.8) | 2 399 234 (67.8) |
| Born abroad | 378 710 (19.1) | 111 611 (6.8) | 462 319 (24.1) | 953 467 (26.9) |
| Unknown | 18 075 (0.9) | 5363 (0.3) | 20 559 (1.1) | 187 859 (5.3) |

*Finnish data only include children born in-country due to incomplete information on migrations.
†Years of follow-up are calculated as first date of death, emigration, turning 18 years of age or last date with available data from the population registry minus the last date of birth, or immigration divided by 365.25.
‡For example, disappeared from register without specification.
§Last date with data available from population registry.

Due to national data protection legislation, country-specific data were stored and analysed in the respective countries using platforms that adhere to country-specific regulations to ensure safe storing and handling of data. Country-specific data were pseudonymised by the registry holders before being transferred to the research team in each country. The common data model allows for the exchange of aggregated or summary data between countries, thus precluding the need to set up separate platforms to exchange data.

## Patient and public involvement

All studies conducted within NONSEnse will be register-based studies only and patients or the public will not be involved in the design or conduct of the planned studies.

## Characteristics of the study population

The national study populations range from 1 637 133 children in Finland to 3 540 560 children in Sweden (table 2). Median follow-up time was 13.1 years in Denmark, 14.2 years in Finland, 12.6 years in Norway and 10.8 years in Sweden. Sweden had the highest proportion of children born abroad; 15.5% compared with 8.4% in Denmark and 11.1% in Norway. The proportion of children who were censored due to migration was lower in Finland, where we only included children born in-country: 0.7% compared with 4.4%–6.2% in the other countries. The lower emigration rate in Finland represents both underreporting due to incomplete information on migration, and a suspected lower risk of moving out of the country for children born in-country, compared with children born abroad. A higher proportion of children without a link to their mother were seen in Sweden; 5.3% compared with 0.2%–1.1% in the other countries. The children without a link to their mother in Sweden were predominantly born abroad (data not presented) and may thus be affected by incomplete registration of migrant families, or children immigrating to Sweden without their mother.

## Exposure assessment: vaccinations across the Nordic countries

Figure 4 depicts the coverage of diphtheria, tetanus and acellular pertussis-containing vaccines, measles-mumps-rubella (MMR) vaccine and rotavirus vaccines for children

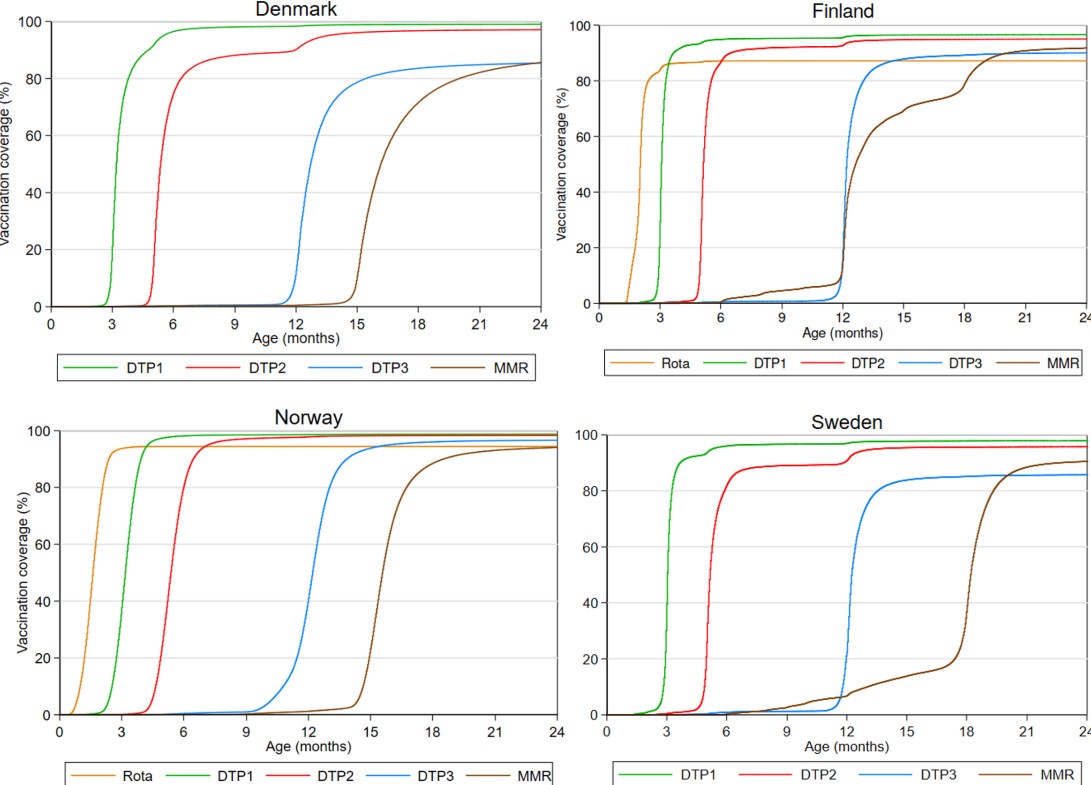

**Figure 4** Vaccination coverage according to age (inverse Kaplan-Meier estimates) among children born in-country in Denmark, Finland, Norway and Sweden. (The coverage reflects the number of registered vaccines and may thus underestimate the actual vaccination coverage in the countries.) The figure inlcudes children born in the respective country from birth cohorts where vaccines administered between 0 and 2 years of age are registered in the vaccination registers (data availability period). The included birth cohorts are 1997–2016 in Denmark; 2009–2015 in Finland; 1995–2016 for DTP and MMR vaccine and 2015–2016 for Rota in Norway; and 2013–2015 in Sweden. The number of children in each birth year is presented in online supplemental appendix 3 sTable 1. DTP1, first dose of diphtheria, tetanus and acellular pertussis-containing vaccine; DTP2, second dose of diphtheria, tetanus and acellular pertussis-containing vaccine; DTP3, third dose of diphtheria, tetanus and acellular pertussis-containing vaccine; MMR, measles-mumps-rubella vaccine; Rota, rotavirus vaccine.

born in each country followed from birth until 2 years of age, date of emigration or date of death, whichever came first (see online supplemental appendix 3 sTable 1 for the coverage at 2 years of age for each of the included birth cohorts in each country).

In Norway, the vaccine uptake rate was highest and closest to the age of recommended vaccination compared with the other countries. In Finland and Sweden, MMR uptake starts at ages earlier than scheduled according to the respective NIPs, which reflects that MMR is recommended to children from 6 and 9 months of age in Finland and Sweden, respectively, before travelling abroad. Although MMR is recommended before travelling abroad in all the Nordic countries, early uptake of MMR is much less frequent in Denmark and Norway which may indicate different interpretation and roll-out of the recommendations. The greater variation in the age at MMR vaccination in Finland reflects different vaccination schedules applied to the included birth cohorts: MMR vaccination was recommended at 14–18 months of age before June 2010 and at 12–18 months (preferably 12 months of age) after June 2010. In Finland, Norway and Sweden, the date of the next vaccination is usually

scheduled during earlier well-baby check-ups or provided by post, whereas in Denmark no formal procedures are in place to ensure timely vaccination, which may explain the different variation in age at vaccination across the countries.

HPV vaccination for girls was introduced in the NIP in 2009 in Denmark, the end of 2013 in Finland, mid-2009 in Norway and in 2012 in Sweden (online supplemental appendix 1). The vaccine is recommended at age 12 years in Denmark, Finland and Norway, and at ages 11–12 years in Sweden. Figure 5 depicts the registered coverage of HPV vaccinations among girls followed from 1 year before the recommended age of vaccination until age 14 years, emigration or death, whichever came first. In Norway, the uptake of the first dose of HPV vaccine follows a steep curve at 12 years of age, representing the age of recommended vaccination (figure 5). The majority of the included birth cohorts in Norway were only able to receive the HPV vaccination free of charge during the school year it was offered, which may have contributed to the high and steep uptake rate. In Sweden, the uptake starts increasing at 11 years of age with a second increase at 12 years of age reflecting that

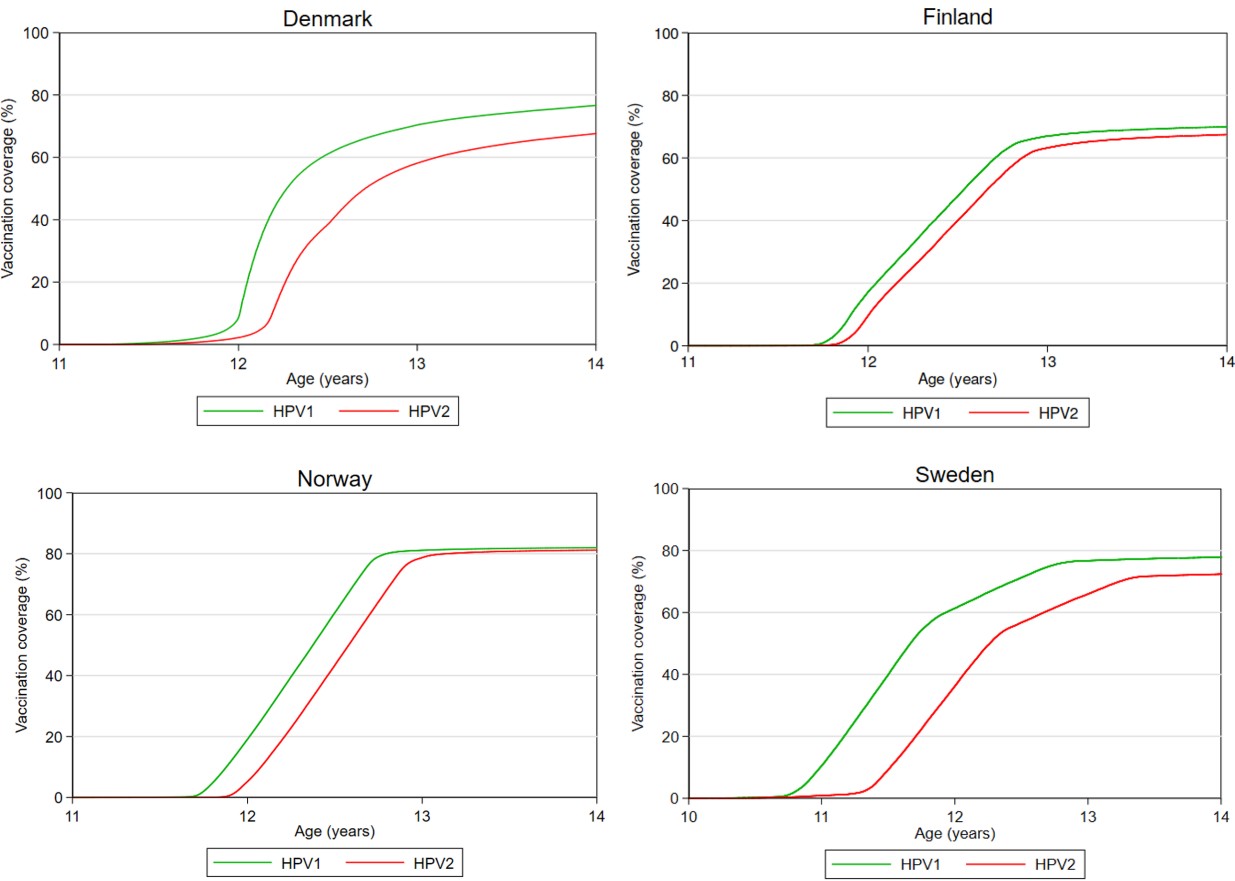

**Figure 5** Human papillomavirus vaccination coverage according to age (inverse Kaplan-Meier estimates) among girls in Denmark, Finland, Norway and Sweden. (The coverage reflects the number of registered vaccinations and may thus underestimate the actual vaccination coverage.) In some countries, the recommended vaccination schedule changed from three to two doses during follow-up. Only the two first doses are reported here. The figure includes girls from birth cohorts where HPV vaccination has been offered. from 1 year before age of recommended vaccination until 14 years of age, and where vaccinations were registered in the vaccination registers. The included birth cohorts are 1998–2004 in Denmark, 2002–2003 in Finland, 1998–2004 in Norway and 2003 in Sweden. The number of girls included in each birth cohort is presented in online supplemental appendix 3 sTable 2. HPV1, first dose of human papillomavirus vaccine; HPV2, second dose of human papillomavirus vaccine.

the vaccine may be administered in either the 5th or 6th grade. In Denmark, uptake starts increasing at 12 years of age corresponding to recommended age of vaccination, but with more variation in the age of vaccination compared with the other countries. The relative low uptake combined with high age variation may be due to vaccination hesitancy following negative media attention from Danish television portraying alleged serious adverse effects of HPV vaccination.[36] Confidence in the safety of the vaccine has since been restored, which is reflected in the slightly increasing vaccination coverage in the last included birth cohort (online supplemental appendix 3 sTable 2). In Finland, the uptake rates follow a straight curve from 12 to 13 years of age followed by a small proportion of children with delayed vaccination. The vaccine uptake at 14 years of age within our cohort was highest in Norway (first dose for the birth cohort 2003: 84.8%), followed by Sweden (77.9%), Finland (69.8%) and Denmark (52.3%) (online supplemental appendix 3 sTable 2).

**Health and sociodemographic characteristics**

Data were available for a different set of years across the Nordic countries. For comparing the study populations in this cohort profile, we only present information from years where data are available in all countries.

*Prescriptions*

Information on redeemed prescriptions was included for the purpose of assessing predefined health outcomes in terms of antibiotic consumption and different atopic outcomes, and to be able to assess potential confounding factors relating to underlying health and healthcare-seeking behaviour. The data legislation regulating access to information on drug utilisation differed across countries. Therefore, data were only obtained for a more narrowly defined subset of ATC codes in Finland and Sweden, compared with Denmark and Norway (online supplemental appendix 3 sTable4). Information from the prescription registries was available from 2005 to 2017 in all countries. We only included information on redeemed

prescriptions with ATC codes available in all countries for the present comparison. The overall proportion of children with redeemed prescriptions ranged from 75.6% in Norway to 86.1% in Finland and varied depending on ATC group (table 3). The proportion of children with redeemed prescriptions in ATC group D 'dermatologicals' was 36.3% in Denmark compared with 20.6%–24.7% in the other countries. Finland had the highest proportion of children with redeemed prescriptions in ATC group J 'antiinfectives for systemic use': 82.3% compared with 62.1%–75.0% in the other countries. In ATC group S 'eye and ear medications', the proportion was lower in Finland (7.4%) compared with the other countries (13.0%–17.9%). For ATC group R 'Respiratory system' and subgroup V01 'Allergens' the proportions were relatively similar across countries.

### Hospital contacts

Information on hospital contacts including inpatient and specialised outpatient care was available in all countries from 2008 to 2016. For comparison across countries, we excluded country-specific codes (eg, codes for health characteristics of newborns in Denmark). The proportion of children with hospital contacts was similar across countries (54.5%–60.2%, table 3). The proportion of children with inpatient contacts ranged from 17.9% in Sweden to 28.7% in Denmark. The proportion of children with outpatient contacts in the patient registers was highest in Sweden (57.5%) and lowest in Denmark (48.8%). The higher proportion of inpatient contacts in Denmark is likely explained by contributions of inpatient contacts without overnight stays, as contacts without overnight stays will predominantly be registered as outpatient contacts in the other countries.[37] The higher proportion of children with outpatient contacts in Sweden may, on the other hand, be explained by a broader set of healthcare facilities (eg, paediatric outpatient clinics) that report to the patient register in Sweden compared with the other countries.[37]

### Birth characteristics

Information on birth characteristics was available for birth cohorts from 1990 to 2016 in all countries (table 4). The completeness of data was high in all countries, ranging from 97.7% to 99.9%. The birth characteristics were also very similar: the median birth weight ranged from 3500 to 3550 g, the proportion of low birthweight (below 2500 g) children ranged from 3.9% to 5.0% and the median gestational age was 40 weeks in all countries. For the variables preterm birth, delivered by caesarean section and singleton births, the proportions only differed by 0.8%–2.7% points across countries. The greatest difference between countries was seen for registration of maternal smoking during pregnancy, which ranged from 8.3% in Norway to 18.2% in Denmark. The proportion with unknown/missing information on maternal smoking ranged from 2.5% in Finland to 45.8% in Norway, which may be explained by the midwives having to inform the mothers of the need for obtaining information on smoking before asking this question in Norway, thus additional effort is required, which may hamper completeness. However, the greater proportion with missing information on maternal smoking in Norway could partly explain the lower proportion with registered maternal smoking during pregnancy, if missing information is more prevalent among smoking mothers.

### Socioeconomic factors

Socioeconomic information is collected yearly in all countries. In the NONSEnse cohort, the information was assessed in the year of birth of each child (table 5) and in the 10th year of life (online supplemental appendix 3 sTable 3). Information from the year of birth was available for the birth cohorts 2004–2015 in all countries. The data presented in table 5 only include children who were born in-country and living in the country throughout their first year of life to ensure that they were present in the country at the time of registration.

In Denmark, 6.2% of the study population had missing information on household income compared with 0%–0.6% in the other countries. We have been unable to identify the reason for the higher proportion in Denmark. The proportion of households with three or more children was 9.4% in Finland compared with 4.4%–5.2% in the remaining Nordic countries. The proportion living with a single parent in the year of birth ranged from 7.9% in Finland to 10.1% in Sweden. Among the remaining socioeconomic variables, the largest cross-country difference was found for the highest attained education of the mother, where information was missing for 21.0% of the children in Sweden compared with 0.2%–3.8% in the other Nordic countries. The proportion of mothers with low education ranged from 11.4% in Sweden to 18.0% in Norway. The high proportion with missing information on maternal education in Sweden is in part caused by a higher proportion of children with an unknown mother in our data set (table 2) but may also be caused by education not being registered for mothers born abroad. Since registration of education is often a necessity for employment in more advanced fields, it is reasonable to assume a higher accuracy for registration of high education as compared with low education.

### Findings to date

The data collection process was completed in March 2022. The findings to date pertain to investigations of similarities and differences in rates of infectious disease hospitalisations[37] and antibiotic consumption.[38] These studies highlight trends in infectious disease morbidity across the Nordic countries and further guide the use of more consistent infectious disease outcome measures for future studies.

The results regarding the non-specific effects of vaccines are at the moment limited to an interrupted time series analysis, which could be undertaken using aggregated data that were ready before all the individual-based

**Table 3** Health characteristics of children present in the respective countries from year 2005 to 2017 (prescriptions) and 2008–2016 (hospital contacts)

| | Denmark | Finland | Norway | Sweden |
|---|---|---|---|---|
| **Prescriptions** | | | | |
| Years of follow-up | 2005–2017 | 2005–2017 | 2005–2017 | 2005–2017 |
| Number of children with follow-up,* n (%) | 1904633 (100.0) | 1634031 (100.0) | 1817231 (100.0) | 3355915 (100.0) |
| Children with redeemed prescriptions,† n (%) | 1592361 (83.6) | 1407548 (86.1) | 1374180 (75.6) | 2542676 (75.8) |
| Mean age during follow-up, mean (SD) | 8.3 (5.2) | 8.3 (5.2) | 8.3 (5.2) | 8.2 (5.3) |
| Prescriptions per child, median (p25–p75) | 4 (1–9) | 5 (2–11) | 3 (1–8) | 3 (1–7) |
| Children with prescriptions with ATC group D, n (%) | 691357 (36.3) | 360910 (22.1) | 449226 (24.7) | 692269 (20.6) |
| Prescriptions per child with ATC group D,†‡§ median (p25–p75) | 1 (1–3) | 1 (1–3) | 1 (1–3) | 1 (1–3) |
| Children with prescriptions with ATC group J, n (%) | 1428652 (75.0) | 1345297 (82.3) | 1129065 (62.1) | 2194753 (65.4) |
| Prescriptions per child with ATC group J,†‡¶ median (p25–p75) | 3 (2–6) | 4 (2–8) | 2 (1–4) | 3 (1–5) |
| Children with prescriptions with ATC group R, n (%) | 806105 (42.3) | 748839 (45.8) | 841066 (46.3) | 1468158 (43.7) |
| Prescriptions per child with ATC group R,†‡** median (p25–p75) | 2 (1–6) | 2 (1–7) | 3 (1–9) | 2 (1–6) |
| Children with prescriptions with ATC group S, n (%) | 248522 (13.0) | 121721 (7.4) | 326077 (17.9) | 521658 (15.5) |
| Prescriptions per child with ATC group S,†‡†† median (p25–p75) | 1 (1–2) | 1 (1–2) | 2 (1–3) | 1 (1–2) |
| Children with prescriptions with ATC group V01, n (%) | 10384 (0.5) | 4662 (0.3) | 11770 (0.6) | 5928 (0.2) |
| Prescriptions per child with ATC group V01,†‡‡‡ median (p25–p75) | 5 (3–9) | 4 (2–7) | 4 (2–8) | 4 (2–8) |
| **Hospital contacts** | | | | |
| Years of follow-up | 2008–2016 | 2008–2016 | 2008–2016 | 2008–2016 |
| Number of children with follow-up,§§ n (%) | 1813600 (100.0) | 1581854 (100.0) | 1738115 (100.0) | 3177371 (100.0) |
| Children with hospital contacts, n (%) | 1069628 (59.0) | 861685 (54.5) | 982808 (56.5) | 1911254 (60.2) |
| Years of follow-up, mean (SD) | 5.8 (3.1) | 5.9 (3.0) | 5.7 (3.0) | 5.5 (3.1) |
| Mean age during follow-up, mean (SD) | 9.2 (5.8) | 9.1 (5.9) | 9.1 (5.8) | 9.0 (5.9) |
| Hospital contacts per child (main diagnosis), median (p25–p75) | 1 (0–2) | 1 (0–4) | 1 (0–3) | 1 (0–4) |
| Children with inpatient contacts, n (%) | 519945 (28.7) | 324292 (20.5) | 420492 (24.2) | 568958 (17.9) |
| Inpatient contacts per child, median (p25–p75) | 1 (1–2) | 1 (1–2) | 1 (1–2) | 1 (1–2) |
| Children with outpatient or emergency room contacts, n (%) | 885243 (48.8) | 839569 (53.1) | 911877 (52.5) | 1826446 (57.5) |
| Outpatient or emergency room contacts per child (1 per day), median (p25–p75) | 2 (1–3) | 3 (1–6) | 2 (1–5) | 3 (1–6) |

Proportions are calculated using number of children with follow-up as the denominator.
*Number of children living in the country at any time in the period 2005–2017.
†Only including ATC subgroups: D02AF, D05, D07, D11, D01, D06, D08, J01–J07, R01, R03, R06, S01G, S03, V01 – thus, not reflecting total use of prescription medicines.
‡Per child redeemed prescriptions of that ATC group.
§ATC group D: dermatologicals.
¶ATC group J: anti-infectives for systemic use.
**ATC group R: respiratory system.
††ATC group S: sensory organs.
‡‡ATC subgroup V01: allergens.
§§Number of children living in the country at any time in the period 2008–2016.
ATC, Anatomical Therapeutic Chemical classification system.

**Table 4** Birth characteristics* of children born in the respective country 1990 to 2016*

| | Denmark | Finland | Norway | Sweden |
|---|---|---|---|---|
| Children born in the respective country from 1990 to 2016 (n) | 1 728 126 | 1 586 526 | 1 591 273 | 2 877 753 |
| Children with information available from the birth registry, n (%) | 1 726 318 (99.9) | 1 576 797 (99.4) | 1 586 895 (99.7) | 2 811 119 (97.7) |
| Birth weight in grams, median (p25–p75) | 3500 (3150–3850) | 3550 (3210–3880) | 3550 (3200–3900) | 3540 (3200–3890) |
| Low birth weight (<2500 g), n (%) | 86 914 (5.0) | 61 546 (3.9) | 73 437 (4.6) | 114 990 (40.) |
| Birth weight missing, n (%) | 23 707 (1.4) | 12 859 (0.8) | 5376 (0.3) | 72 892 (2.5) |
| Gestational age in weeks, median (p25–p75) | 40 (39–41) | 40 (39–40) | 40 (39–41) | 40 (39–40) |
| Preterm birth, n (%) | 107 656 (6.2) | 85 069 (5.4) | 98 923 (6.2) | 163 168 (5.7) |
| Gestational age missing, n (%) | 29 250 (1.7) | 16 083 (1.0) | 59 264 (3.7) | 68 973 (2.4) |
| Delivered by caesarean section, n (%) | 305 738 (17.7) | 258 261 (16.3) | 238 013 (15.0) | 435 680 (15.1) |
| Mode of delivery missing, n (%) | 1808 (0.1) | 9729 (0.6) | 4378 (0.3) | 66 634 (2.3) |
| Singleton, n (%) | 1 660 213 (96.1) | 1 531 748 (96.5) | 1 535 556 (96.5) | 2 731 980 (94.9) |
| Child order including the child itself, n (%) | | | | |
| 1 (firstborn) | 743 923 (43.0) | 647 134 (40.8) | 658 877 (41.4) | 1 211 084 (42.1) |
| 2 | 635 849 (36.8) | 532 868 (33.6) | 568 765 (35.7) | 1 023 228 (35.6) |
| 3 | 243 162 (14.1) | 244 137 (15.4) | 257 294 (16.2) | 398 331 (13.8) |
| 4 or more | 86 090 (5.0) | 149 327 (9.4) | 101 959 (6.4) | 178 184 (6.2) |
| Missing | 19 102 (1.1) | 13 060 (0.8) | 4378 (0.3) | 66 926 (2.3) |
| Maternal smoking during pregnancy, n (%) | 314 174 (18.2) | 238 337 (15.0) | 132 734 (8.3) | 310 691 (10.8) |
| Maternal smoking unknown, n (%) | 134 332 (7.8) | 39 277 (2.5) | 728 038 (45.8) | 143 529 (5.0) |

*Information, including percentages, is reported according to the number of children born in-country from 1990 to 2016.

**Table 5** Socioeconomic factors at birth for children born in the respective country 2004 to 2015

| | Denmark | | Finland | | Norway | | Sweden | |
|---|---|---|---|---|---|---|---|---|
| | n | % | n | % | n | % | n | % |
| Children present in-country at birth from 2004 to 2015 | 729294 | | 699052 | | 706443 | | 1314701 | |
| Birth cohorts included | 2004–2015 | | 2004–2015 | | 2004–2015 | | 2004–2015 | |
| Income quintile at birth | | | | | | | | |
| First (lowest) | 134634 | 18.5 | 138965 | 19.9 | 137551 | 19.5 | 247237 | 18.8 |
| Second | 137041 | 18.8 | 138997 | 19.9 | 141566 | 20.1 | 265557 | 20.2 |
| Third | 137390 | 18.8 | 139012 | 19.9 | 141962 | 20.1 | 267347 | 20.3 |
| Fourth | 137415 | 18.9 | 138998 | 19.9 | 141995 | 20.1 | 267349 | 20.3 |
| Fifth | 136935 | 18.8 | 138900 | 19.9 | 141533 | 20.1 | 266528 | 20.3 |
| Unknown | 45531 | 6.2 | 4180 | 0.6 | 644 | 0.1 | 605 | 0.0 |
| Number of children in the household the year the child is born | | | | | | | | |
| 1 | 310237 | 42.6 | 287312 | 41.1 | 298563 | 42.3 | 574229 | 43.7 |
| 2 | 278396 | 38.2 | 237291 | 33.9 | 263726 | 37.4 | 487446 | 37.1 |
| 3 | 106184 | 14.6 | 104278 | 14.9 | 108822 | 15.4 | 176338 | 13.4 |
| >3 | 32106 | 4.4 | 65527 | 9.4 | 33496 | 4.7 | 68060 | 5.2 |
| Unknown | 2023 | 0.3 | 4644 | 0.7 | 644 | 0.1 | 605 | 0.0 |
| Single parenthood in the years the child is born | | | | | | | | |
| Yes | 58646 | 8.0 | 55089 | 7.9 | 68018 | 9.6 | 132243 | 10.1 |
| No | 668277 | 91.7 | 639319 | 91.5 | 635689 | 90.1 | 1181775 | 89.9 |
| Unknown | 2023 | 0.3 | 4644 | 0.7 | 1544 | 0.2 | 605 | 0.0 |
| Highest attained educational level* of the mother on the date the child is born | | | | | | | | |
| Low education | 114880 | 15.8 | 98608 | 14.1 | 126777 | 18.0 | 149673 | 11.4 |
| Medium education | 261761 | 35.9 | 279687 | 40.0 | 201316 | 28.5 | 431880 | 32.9 |
| High education | 336536 | 46.2 | 319530 | 45.7 | 350684 | 49.7 | 457040 | 34.8 |
| Unknown | 15769 | 2.2 | 1227 | 0.2 | 26474 | 3.8 | 276030 | 21.0 |

*Highest attained education was categorised based on the International Standard Classification of Education (ISCED) 2011 using the main groups.[53]

data were obtained in all countries.[39] Future studies will include population-level investigations of natural experiments in the form of introduction of new vaccines or changes in the immunisation programmes, as well as individual-level studies comparing vaccinated and unvaccinated children with a given vaccine using multiple different study designs.

## FURTHER DETAILS
### Strengths and limitations
The NONSEnse project represents a unique undertaking for conducting register-based epidemiological studies of the overall health effects of routine childhood vaccines.

Data are stored separately in each country, which prevents conducting analyses on the joint data, which is a limitation of the project. However, the common data model enables analysis plans and statistical code to be written in one country and sent to the other countries that can then perform the same analyses and share the results (figure 3). The use of a common data model thus minimises the risk that different country-specific analytical decisions will hinder comparability of results.

The use of register data presents both strengths and weaknesses. A strength pertains to the multitude of information available for the entire study population and linked to the individual, which minimises selection bias and enables cohort studies with prospective follow-up and control for multiple confounding factors. The generalisability of the Finnish cohort is limited to children born in-country. However, for most of studies to be undertaken within this project, this will have limited implications since we will often restrict the study population to children born in-country for the studies of childhood vaccinations to ensure complete information on vaccinations given from birth. Limitations include that not all the wished-for information is available in all countries

and registration may be incomplete, which limits the possibility to, for example, adjust for hypothesised confounding factors such as day care attendance and lifestyle factors. Also, previous studies[2] have found the non-specific effect of a vaccine to be strongest when it is the most recent vaccine administered. Therefore, it is relevant to include information on vaccines other than the ones offered through the NIP. In Denmark, Finland and Norway, vaccines outside the NIP may also be registered in the vaccination registers, but registration of these vaccines has only been mandatory in more recent years.[26 27 30] In Sweden, only vaccinations within the NIP are included in the vaccination register. The analyses are thus limited by different possibilities to assess the effect of a given vaccine as long as it is the most recent vaccine, both within and across countries.

In all the Nordic countries, information on emigration relies on the individual reporting resettlement to the authorities. This is mandatory when leaving the country for more than 6 months in Denmark[40] and Norway,[41] and for more than 12 months in Sweden[21] and Finland.[42] Thus, incomplete information on emigrations, due to leaving the country for shorter periods of time or if parents fail to register the resettlement, may result in children being lost to follow-up without us knowing it from the registers. This may in turn result in our studies underestimating events, for example, infectious disease hospitalisations, as these are only registered for children who are in the country.

Overall, it is clear that expert knowledge is needed before combining and using Nordic register data for research purposes.[20] As such, an important strength of NONSEnse pertains to the data harmonisation process through biweekly analysis workshops involving designated research groups from each of the four countries with expert knowledge on country-specific register data, the healthcare systems and immunisation programmes.

### Validity of exposure and outcome measures

In all countries, the vaccines offered through the NIP are subject to mandatory registration. However, validity depends on the reporting accuracy by the healthcare providers who administer the vaccinations. A Danish study validated the coverage of MMR from the registers using medical records from the general practitioner in a subset of the population and found that the coverage in the register was 86% compared with 94% through inspection of the medical records.[43] A similar comparison conducted in Sweden also found under-reporting of MMR in the register of around 5–7 percentage units (unpublished). It is unlikely that under-reporting of vaccines is associated with the outcomes investigated within the NONSEnse project; therefore, the misclassification will most likely be non-differential and would thus bias the results towards no association.

The prescription registers only contain information on drugs dispensed from filled prescriptions, whereas some drugs are also available over the counter, which are not included in the registers. This includes, for example, weak corticosteroids for topical use (ATC: D07AA) or drugs used to treat symptoms in the eye due to, for example, allergy (ATC: S01G). It is thus possible that the observed cross-country differences in the proportion of children with these prescriptions are affected by national policies or guidelines, or the behaviour of the prescriber or purchaser. Atopic outcomes will, in part, be identified using filled prescriptions for products that are also available over the counter, which may hamper cross-country comparability. Antibiotics, however, are prescription drugs in all four countries and thus not affected by over-the-counter purchases.

Several differences in healthcare organisation, administration and registration may hamper cross-country comparability of the health outcomes included in this project. A strength of NONSEnse is the thorough investigation of the intended outcomes in independent studies which has informed and maximised comparability of the outcome measures to be used in the subsequent studies of non-specific effects of vaccines.

### Methodological considerations

Evaluating the effect of implemented vaccination programmes is challenging; the high vaccine uptake rate makes comparisons between vaccinated and unvaccinated children difficult due to the individual factors that determine vaccine uptake. Healthy vaccinee bias may arise if the healthiest children are more likely to follow the vaccination recommendations than the less healthy children.[44] However, due to different vaccination schedules in different countries, the children who have received MMR at, for example, 15 months of age may be classified as vaccinated according to schedule, too early or too late, depending on the country. Furthermore, age is a strong predictor of both vaccination and the risk of infectious diseases.[37] A strength therefore pertains to the observed delay in age at vaccination within each country, which facilitates comparison of different vaccination statuses among children of the same age. For vaccines with a steep and high uptake at the recommended age of vaccination, the children who do not receive the vaccines as scheduled are more likely a selected subgroup of the population, thus hampering comparability with the rest of the population. In contrast, larger variation in the age at vaccination increases comparability between children with different vaccination status according to age.

A strength of this study set-up is the many differences in the immunisation programmes, and in changes to the immunisation programmes, the country-specific bias structures and the possibility to integrate results from different study designs, which facilitate triangulation that can strengthen the potential for making causal

deductions.[12 13] The project has already led to useful new information regarding differences and similarities in childhood morbidity between the Nordic countries. Most importantly, the project will increase our understanding of vaccines and how they may affect health in more general ways—holding potential for direct translation into more efficient immunisation programmes and improved child health.

**Author affiliations**
¹Bandim Health Project, Research Unit OPEN, Department of Clinical Research, University of Southern Denmark, Odense, Denmark
²Danish Institute for Advanced Study, University of Southern Denmark, Odense, Denmark
³Division of Infection Control, Norwegian Institute of Public Health, Oslo, Norway
⁴Department of Public Health Analysis and Data Management, Public Health Agency of Sweden, Solna, Sweden
⁵Department of Public Health and Welfare, Finnish Institute for Health and Welfare, Helsinki, Finland
⁶Department of Information Services, Finnish Institute for Health and Welfare, Helsinki, Finland
⁷Department of Clinical Epidemiology, Aarhus University Hospital, Aarhus, Denmark

**Contributors** LG, IL, HE, HN, BF, ML, AAP, LT, CSB and SS conceptualised the manuscript. LG, IL, HE, HN and SS designed the methodology used to present the included data. LG, IL, HE and ML managed the data curation, undertook the country-specific coding and produced the aggregated data presented. LG, IL, HE, HN, BF, ML, AAP, LT, CSB and SS investigated and validated the aggregated data produced. LG and IL did the formal analysis of data. LG, IL, HE, BF, LT, HN, AAP and SS obtained the data to be included in this cohort. LG drafted the first version of the manuscript. IL, HE, HN, BF, ML, AAP, LT, CSB and SS critically revised the draft. LG and IL produced the visualisations. CSB and SS supervised the project. LG, IL, HN, BF, ML, AAP, LT, CSB and SS obtained the funding for the present project. LG is the guarantor of the study.

**Funding** This work was supported by NordForsk (grant number: 83839), Odense University Hospital Research Fund (A-number: 2519) and the faculty scholarship from the University of Southern Denmark.

**Competing interests** AAP, HN and ML are investigators in vaccine-related studies for which THL has received funding from GSK, Pfizer and Sanofi Pasteur.

**Patient and public involvement** Patients and/or the public were not involved in the design, or conduct, or reporting, or dissemination plans of this research.

**Patient consent for publication** Not applicable.

**Provenance and peer review** Not commissioned; externally peer reviewed.

**Data availability statement** Data may be obtained from a third party and are not publicly available. Due to data protection rules and ethical permissions, we are not allowed to share the individual-level data. However, the insights presented in this cohort profile, including the common data model, can serve to guide the construction of similar Nordic databases by other researchers fulfilling the requirements to obtain Nordic registry data. The possibility to generate Nordic population-based cohorts could, for example, be used to study health interventions and outcomes related to the SARS-CoV-2 pandemic across the Nordic countries.

**ORCID iDs**
Lise Gehrt http://orcid.org/0000-0002-6590-6263
Hélène Englund http://orcid.org/0000-0002-6258-7818
Signe Sørup http://orcid.org/0000-0003-1942-683X

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
