## [Reviewer comments · BMJ Open]

ARTICLE DETAILS

TITLE (PROVISIONAL)	Cohort Profile: Childhood morbidity and potential non-specific effects of the childhood vaccination programmes in the Nordic countries (NONSEnse): Register-based cohort of children born 1990-2017/2018.
AUTHORS	Gehrt, Lise; Laake, Ida; Englund, Hélène; Nieminen, Heta; Feiring, Berit; Lahdenkari, Mika; Palmu, Arto A.; Trogstad, Lill; Benn, Christine Stabell; Sørup, Signe

VERSION 1 – REVIEW

REVIEWER	Abtahi , Shahab Universiteit Utrecht, Division of Pharmacoepidemiology and Clinical Pharmacology, Utrecht Institute for Pharmaceutical Sciences
REVIEW RETURNED	13-Oct-2022

GENERAL COMMENTS	I read with interest the study by Gehrt et al on "Cohort Profile: Childhood morbidity and potential non-specific effects of the childhood vaccination programmes in the Nordic countries (NONSEnse): Register-based cohort of children born 1990-2017/2018." The 'Cohort profile' described in this study would be a great addition to the BMJ Open, and it already covers the overall direction of the cohort and future studies, funding sources, and data access considerations. However, it may benefit from the following (major) questions/ remarks: 1- As I understood from the title and objectives of this cohort profile, the COVID-19 pandemic period between 2019-2022 and the SARS-CoV-2 vaccines are not included in this setting. Can authors explain why such decision has been made, and why you are not going to design and conduct studies using he same infrastructure from the 4 countries on non-specific effects of SARS-CoV-2 vaccination? 2- Following my point above, is there any plan by this team to address and evaluate an imminent public health or clinical intervention using the data collected in this cohort? 3- In Abstract, Findings to date, I think it would be better to focus on what you have done to make this cohort and its attributes and characteristic instead of going into details of the studies that you conducted based on this cohort, without mentioning the results of those studies in the main text of this paper. You may consider to rephrase this part. Otherwise, if you want to include those studies inside this 'Cohort profile' as some practical examples, consider to include a brief summary of them (design, main results, conclusions) also in the main text of this paper.
--

	4- Are all children living in these 4 countries included in the health registers of the respecting country? In Abstract, you mention only "permanent residents" will be included, where it excludes all other residents, such as temporary residents, refugees, other immigrants, students, etc. I can imagine, this will lead to selection bias, and reduces the inclusiveness and generalisability of your data then?! What is your plan to overcome this? As I noted in the paper, this issue is more relevant for Finland. 5- It is unclear from the Setting, page 8, lines 21-41, which national registers or sub-databases have been linked and included from each country under the broader umbrella term of "national registers"? As I know from the Denmark example, there are various registers for hospitalisation records & outpatient diagnoses (DNPR), outpatient pharmacy dispensings (Register of Medicinal Product Statistics), speciality clinical registers (such as DANBIO), etc. It will be beneficial if authors elaborate more on this in this section. 6- In Table 2, I was concerned about the length of follow-up. First, as this variable is not expected to have a normal distribution, I advise to report it with a median and interquartile range instead of mean and standard deviation. Part of your included individuals (born between 1990 and 2001, roughly 43%) had a chance to finish the study period at the age of 18 (if born inside the countries and not immigrated at an older age), and 57% reached the end of study (2018) before turning 18. The median value should more or less correspond to this simple estimate (something between 12 and 13 years). This is important, also as good quality check, to see how you were able to retrieve data from various registers.. 7- Once I read that "the Finnish Prescription Center started gradually in 2010 and collects all redeemed prescriptions irrespective of reimbursement. By 2017, practically all prescriptions were included in the Finnish Prescription Center." So how was it possible that authors state in page 14: "Information on redeemed prescriptions was available from 2005 to 2017 in all countries." can you explain this discrepancy? In case of incomplete drug utilisation data from Finland for several years of this cohort's life, I am seriously concerned about the possibility of conducting many of the foreseen studies including Finnish data.. 8- As data privacy rules at each national centre are understandable, I would not call this a limitation if it is not possible to aggregate data at one place and conduct analyses on it. Using a Common Data Model helps to harmonise the data tables and build similar analytic datasets, on which once can run the very same analytic scripts, with the possibility of pooling the estimates at the final stage with a meta-analysis and produce a pooled cumulative risk estimate from all centres involved, with more generalisability and representativeness. I would call this a strength not a limitation! 9- Authors already mentioned the funding information for this cohort, but the information on its maintenance and data management is lacking. Please provide this information too. Is there any secure platform for handling of data, curing datasets, running the analyses, saving and spreading the study results, and
--	--

	eventual pooling of results from centres? How the authors planned to distribute the common analytic script? Which statistical platform is usually used by teams? 10- Authors mentioned a 'Data sharing statement'. However, any remarks on future collaboration with other investigators from other countries using the same data from this cohort or adding it to other data from other data sources is lacking. Starting new collaborations and bringing together different cohorts and data is supposed to be one goal of publishing such cohort profiles. Please elaborate more on this.
--	--

REVIEWER	Latipov, R Research Institute of Virology
REVIEW RETURNED	20-Oct-2022

GENERAL COMMENTS	The presented paper is not clearly a study but a presentation of the Cohort Profile. Further, it could be used as an annex to the main study paper. In case a journal accepts such publications, it could be accepted for publication. Well-explained cohorts (huge number of included children), assessing factors (limited number but all available) and assessment model. Clearly presented limitations.
---

REVIEWER	Zirimenya , Ludoviko MRC/UVRI and LSHTM Uganda Research Unit, Immunomodulation and Vaccines Programme
REVIEW RETURNED	25-Oct-2022

GENERAL COMMENTS	Generally, this is an exciting area of focus and the researchers have invested a lot of time, resources, and energy to undertake the study. However, this manuscript is not well structured so it is difficult to understand. The abstract is not well structured, the author is referred to the journal's author submission guidelines that clearly show how it should be structured. The researchers should use the IMRAD basic structure of scientific papers. The introduction section is well written but unfortunately the methods, results, and discussion sections, need to be improved. For example, the section on cohort description covers both description of the methods and results, these should be separate. When I read the abstract, I thought that the manuscript describes a protocol of planned work, but on further reading, they share some findings one wonders how they are linked to non-specific effects of childhood vaccinations. Clarity is needed on what exactly this manuscript is intended to be. Please be aware of the tense so that it does not read as if this is a protocol paper. E.g. We will analyse.... Is better phrased as we analyzed etc. The main aim is clearly mentioned but what are those specific research questions mentioned in line 53 that guided the analysis? As per the STROBE guidelines, the methods section can be better described to make the methods more understandable. Please
--

	include items such as possible sources of bias and how they were addressed, statistical methods used, etc. such aspects are missing and will help a reader understand the paper better. Furthermore, the structure of the manuscript should be written to align with the STROBE guidelines. E.g. the discussion of the results is missing to highlight how the results were interpreted and their generalizability etc. Please include a supplementary report of its checklist. Prior to the start, where the criteria set out of who to include and exclude from the study to avoid any bias? As well description of methods of follow-up should be shared. Were any ethical approvals obtained for the study in all the countries from regulatory and ethical committees? How was access to data granted and confidentiality maintained?
--	--

REVIEWER	Pinot de Moira , Angela University of Copenhagen
REVIEW RETURNED	31-Oct-2022

GENERAL COMMENTS	Cohort Profile: Childhood morbidity and potential nonspecific effects of the childhood vaccination programmes in the Nordic countries (NONSEnse): Register-based cohort of children born 1990-2017/2018. The paper describes the creation of a Nordic register-based cohort to investigate the non-specific effects of vaccines on the overall health of children. To create the cohort, health and sociodemographic register data from 9,072,420 children aged 0-17 years born between 1990 and 2018 and registered as living in Denmark, Finland, Norway or Sweden were harmonized according to a Common Data Model. The cohort will be a useful resource for examining the effects of vaccines on children's health and the paper is well written. However, I have some recommendations for improvement before the paper is published. Major Comments 1) Although the aims of the NONSEnse project are clearly described, it would be useful to also set out the aims of the paper. I assume these are to describe how the register-based cohort was created, the characteristics of the cohort and to give an overview of the data included. 2) The Lexis diagrams are very helpful in obtaining an overview of available data, but in addition to these, it would be useful if the authors could explain the rationale for their chosen start date (1990) of follow-up. Vaccination data are only available from 1995 in Norway, later in the other countries and in Sweden no vaccination data were available before 2013. Since children were only followed until 18 years, children born 1990-1995 in Sweden will have no vaccination data. Similarly, hospital contacts have only reached national coverage since 2008 in Norway, meaning that many children will be missing these data. I wonder if a later start date would have offered more consistency across the countries.
--

	3) More information on the harmonization process, the Common Data Model and the selection of variables in the main section of the manuscript would be useful. Also, for comparability across countries, it would be useful if the tables included in the “NONSense Common Data Model” document could be combined for the four countries (i.e. one table for prescriptions, one table for hospital contacts etc.). Minor comments 1) In the abstract, under “Findings to date”, the authors state: “we have identified indicators measuring similar levels of infectious disease morbidity across these settings. We have also conducted an interrupted-time series analysis of the association between the second dose of measles-mumps-rubella (MMR) vaccination and the rate of infectious disease hospitalisations due to infections not targeted by the MMR vaccine”, however, these findings are presented in other papers and only discussed in the current paper. 2) I would rephrase the aim on lines 45-47 of the introduction. 3) Page 8, line 49: why the different follow-up end date in the four countries? 4) Page 10, line 50: Finland to (rather than Finlandto). 5) Page 18, line 36: although given in Table 4, it would be useful to also clarify the definition of low birthweight in the text. 6) Could the large amount of missing data on maternal smoking in Norway also partly explain the lower prevalence of maternal smoking in this country? 7) Page 25: am I right in understanding that these data are not open to other researchers to utilise, even for researchers based within the Nordic countries?
--	---

REVIEWER	Dey, Aditi National Centre for Immunisation Research and Surveillance of Vaccine Preventable Diseases
REVIEW RETURNED	08-Nov-2022

GENERAL COMMENTS	Thanks for giving me the opportunity to review the manuscript titled, “Cohort Profile: Childhood morbidity and potential non-specific effects of the childhood vaccination programmes in the Nordic countries (NONSense): Register-based cohort of children born 1990-2017/2018”. I commend the authors for undertaking this important piece of research. However, the manuscript appears very long and had gaps in information on the non-specific effects of the childhood vaccination programmes. I would have preferred to see what were the outcome measures (non-specific effects) that the researchers were investigating. How were these non-specific defined/ascertained/measured? How long was the follow-up period for each of these non-specific effects to develop and or resolve? How were confounders accounted for? The authors mention in the abstract about undertaking a time-series analysis of the association between the second dose of measles-mumps-rubella (MMR) vaccination and the rate of infectious disease hospitalisations due to infections not targeted by the MMR vaccine. Again, I was unable to find these results in the main body of the manuscript. Ethics approval processes were not mentioned in the manuscript though the authors mentioned that due to national data protection legislation, country-specific data were stored and analysed in the respective countries.
---

VERSION 1 – AUTHOR RESPONSE

Reviewer: 1

Dr. Shahab Abtahi , Universiteit Utrecht

Comments to the Author:

I read with interest the study by Gehrt et al on "Cohort Profile: Childhood morbidity and potential non-specific effects of the childhood vaccination programmes in the Nordic countries (NONSEnse): Register-based cohort of children born 1990-2017/2018." The 'Cohort profile' described in this study would be a great addition to the BMJ Open, and it already covers the overall direction of the cohort and future studies, funding sources, and data access considerations. However, it may benefit from the following (major) questions/ remarks:

Response 2: We thank the reviewer for the positive view and comments to improve the manuscript

1- As I understood from the title and objectives of this cohort profile, the COVID-19 pandemic period between 2019-2022 and the SARS-CoV-2 vaccines are not included in this setting. Can authors explain why such decision has been made, and why you are not going to design and conduct studies using the same infrastructure from the 4 countries on non-specific effects of SARS-CoV-2 vaccination?

Response 3: We thank the reviewer for the very relevant question. When conceptualizing the cohort, SARS-CoV-2 had not yet emerged. The present cohort includes data until 2017-2018 only, thus we do not have data from years with SARS-CoV-2. The platform presented, including the common data model could however be used when applying for new data to create a similar data structure to study SARS-CoV-2 related vaccines and outcomes across the Nordic countries. We have further added this statement to the data sharing statement as an example of future research that can emerge from this type of cohorts. Please also see Response 4.

2- Following my point above, is there any plan by this team to address and evaluate an imminent public health or clinical intervention using the data collected in this cohort?

Response 4: We thank the reviewer for the interest in our continued studies. While the data structure would be appropriate to undertake such studies, our data is unfortunately not continuously updated, thus we will not have the data at hand for analysing an imminent public health or clinical intervention. Investigating imminent public health interventions is beyond the aim of the NONSEnse project and would further require new ethical approvals to use the data for other purposes than the original purpose. While such studies are beyond the scope of the present project, the insights on data structure and common data model could be used by other researchers to create a similar setup across the Nordic countries to undertake such studies. We have updated our data sharing statement, to illustrate some of the research potential for future projects using Nordic registry data, now reading "Due to data protection rules and ethical permissions, we are not allowed to share the individual-level data. However, the insights presented in this cohort profile, including the common data model, can serve to guide the construction of similar Nordic databases by other researchers fulfilling the requirements to obtain Nordic registry data. The possibility to generate Nordic population-based cohorts could for example be used to study health interventions and outcomes related to the SARS-CoV-2 pandemic across the Nordic countries. "

3- In Abstract, Findings to date, I think it would be better to focus on what you have done to make this cohort and its attributes and characteristic instead of going into details of the studies that you conducted based on this cohort, without mentioning the results of those studies in the main text of this

paper. You may consider to rephrase this part. Otherwise, if you want to include those studies inside this 'Cohort profile' as some practical examples, consider to include a brief summary of them (design, main results, conclusions) also in the main text of this paper.

Response 5: We thank the reviewer for the comment. We have included this paragraph in the abstract in accordance with the author guidelines for cohort profiles "Abstract Use these headings to provide brief descriptions of the following /.../ Findings to date: what data has been collected so far and any major results /.../". We realize that the description of the required content of this section could also be fulfilled without mentioning the specific studies that have been undertaken so far. We have thus updated the section in the main text to focus on the status of collecting the data (complete), and furthermore state that the findings to date pertain to 1) the initial investigations of antibiotic consumption and hospital contacts for infections across the Nordic countries, and 2) results from studies of nonspecific effects is limited to an interrupted timeseries analysis, which could be undertaken using aggregated data that was ready before all the individual based data was obtained in all countries. The findings to date section in the abstract has been updated to read "Data collection and harmonisation according to a Common Data Model was completed in March 2022. As a prerequisite for comparing the effects of childhood vaccinations on the overall health of children across the Nordic countries, we have identified indicators measuring similar levels of infectious disease morbidity across these settings. So far studies pertaining to non-specific effects of vaccines are limited to investigations that could be undertaken using aggregated datasets that were available before the NONSENSE cohort with individual level information was completely setup."

4- Are all children living in these 4 countries included in the health registers of the respecting country? In Abstract, you mention only "permanent residents" will be included, where it excludes all other residents, such as temporary residents, refugees, other immigrants, students, etc. I can imagine, this will lead to selection bias, and reduces the inclusiveness and generalisability of your data then?! What is your plan to overcome this? As I noted in the paper, this issue is more relevant for Finland.

Response 6: The cohorts include all children born in the countries and all children who became permanent residents after migration to the countries. Temporary residents, e.g., students, are not expected to represent a large group among children below 18 years of age. Thus, we do not expect this restriction to result in selection bias. We have updated the first sentence in the description of the study population to state "We used national population registers to identify all children aged 0-17 years, who were born or became permanent residents after migrating to one of the Nordic countries..."

In Finland, however, we could only include children born in Finland, as also mentioned in the presentation "The population data obtained in Finland had incomplete information on migration history before 2014 and thus we were unable to assess the date of entering the country for children born abroad. As a result, we limited the Finnish study population to children born in the country to ensure that they were present in the country from the beginning of follow-up". This will affect the generalisability of the results obtained from Finland. In practice, however, this will have limited implications for the studies of non-specific effects, as we will often need to restrict the cohort to children born in-country to have full information on all vaccines given in infancy. We have added this description to the strengths and limitations section that now includes "The generalisability of the Finnish cohort is limited to children born in-country. However, for most of studies to be undertaken within this project, this will have limited implications since we will often restrict the study population to children born in-country for the studies of childhood vaccinations to ensure complete information on vaccinations given from birth."

5- It is unclear from the Setting, page 8, lines 21-41, which national registers or sub-databases have been linked and included from each country under the broader umbrella term of "national

registers"? As I know from the Denmark example, there are various registers for hospitalisation records & outpatient diagnoses (DNPR), outpatient pharmacy dispensings (Register of Medicinal Product Statistics), speciality clinical registers (such as DANBIO), etc. It will be beneficial if authors elaborate more on this in this section.

Response 7: We thank the reviewer for highlighting that the data sources are not clearly presented. The paragraph referred to by the reviewer pertains to a general description of the availability of registry data across the Nordic Countries. We have included the specific registries included in each of the Nordic countries in figure 2 in the description on the next page "source and content of data". The common data model further presents the specific tables we have used from each of the included registries. In the main text, we provide a more elaborate description of the vaccination registries included in each country, and for the main outcome we specifically mention the registries used for outcome measures i.e patient registries and prescriptions registries. In addition to the specific names of vaccination registries and prescription registries included, we have now also added the specific names for the patient registries.

We have omitted the names of each of the registries for all the included covariates in the main text due to space considerations, as we with many different registries from 4 different countries would be listing a large number of registries. The exact registries can be found in Figure 2.

If the editor wishes, we can further include these in the main text.

6- In Table 2, I was concerned about the length of follow-up. First, as this variable is not expected to have a normal distribution, I advise to report it with a median and interquartile range instead of mean and standard deviation. Part of your included individuals (born between 1990 and 2001, roughly 43%) had a chance to finish the study period at the age of 18 (if born inside the countries and not immigrated at an older age), and 57% reached the end of study (2018) before turning 18. The median value should more or less correspond to this simple estimate (something between 12 and 13 years). This is important, also as good quality check, to see how you were able to retrieve data from various registers..

Response 8: We thank the reviewer for this suggested change. We have now presented median follow-up instead of mean.

7- Once I read that "the Finnish Prescription Center started gradually in 2010 and collects all redeemed prescriptions irrespective of reimbursement. By 2017, practically all prescriptions were included in the Finnish Prescription Center." So how was it possible that authors state in page 14: "Information on redeemed prescriptions was available from 2005 to 2017 in all countries." can you explain this discrepancy? In case of incomplete drug utilisation data from Finland for several years of this cohort's life, I am seriously concerned about the possibility of conducting many of the foreseen studies including Finnish data.

Response 9: We thank the reviewer for highlighting this inconsistency in description. We have added the following text to the description of the source of the Finnish prescription data: "We combined the information from the Finnish Prescription Center and the Finnish Benefits Registry to obtain the most complete information on redeemed prescriptions (see Appendix 2 for details on source of data)". We have further updated the description page 14 to "Information from the prescription registries was available from 2005 to 2017 in all countries". We further agree to the limitations of using the Finnish prescription data, especially for some years and specific products, which is also part of the importance of publishing this cohort profile, so other researchers will pay attention to this limitation when undertaking future studies.

8- As data privacy rules at each national centre are understandable, I would not call this a limitation if it is not possible to aggregate data at one place and conduct analyses on it. Using a

Common Data Model helps to harmonise the data tables and build similar analytic datasets, on which once can run the very same analytic scripts, with the possibility of pooling the estimates at the final stage with a meta-analysis and produce a pooled cumulative risk estimate from all centres involved, with more generalisability and representativeness. I would call this a strength not a limitation!

Response 10: We are happy that the reviewer appreciates the great potential of using a common data model for investigations across settings. The limitation was included to demonstrate the need for a common data model when data is located separately in each country.

9- Authors already mentioned the funding information for this cohort, but the information on its maintenance and data management is lacking. Please provide this information too. Is there any secure platform for handling of data, curing datasets, running the analyses, saving and spreading the study results, and eventual pooling of results from centres? How the authors planned to distribute the common analytic script? Which statistical platform is usually used by teams?

Response 11: Thank you for highlighting the lacking information. In the section on source and content of data we state "Due to national data protection legislation, country-specific data were stored and analysed in the respective countries." We have now added the following "using platforms that adhere to country specific regulations to ensure safe storing and handling of data. Country specific data was pseudonymized by the registry holders before being transferred to the research team in each country. The common data model allows for the exchange of aggregated or summary data between countries, thus precluding the need to set up separate platforms to exchange data."

10- Authors mentioned a 'Data sharing statement'. However, any remarks on future collaboration with other investigators from other countries using the same data from this cohort or adding it to other data from other data sources is lacking. Starting new collaborations and bringing together different cohorts and data is supposed to be one goal of publishing such cohort profiles. Please elaborate more on this.

Response 12: As stated in the data sharing statement, it is unfortunately not possible to share individual level information due to data legislation. Meanwhile it is possible for other researchers fulfilling the requirements to obtain the same data from the registries. We have elaborated on the data sharing statement, please see Response 4.

Reviewer: 2

Dr. R Latipov, Research Institute of Virology Comments to the Author:

The presented paper is not clearly a study but a presentation of the Cohort Profile. Further, it could be used as an annex to the main study paper. In case a journal accepts such publications, it could be accepted for publication. Well-explained cohorts (huge number of included children), assessing factors (limited number but all available) and assessment model. Clearly presented limitations.

Response 13: We thank the reviewer for the comment and are happy that you find the descriptions well explained. Indeed, BMJ Open accepts "Cohort Profile" articles. You are correct when stating that this manuscript does not present a study, but rather a presentation of the cohort, which will be used to undertake multiple studies. We have added the aim of the cohort profile to the introduction to make this more clear, in line with the suggestion from reviewer 4, please see Response 24: "The aim of the present cohort profile is to describe the content and quality of the data included in the registry-based NONSEnse cohort and present characteristics of the cohort, thereby demonstrating the research potential of the NONSEnse cohort. The insights presented can be used to guide future epidemiological research projects using registry data from the Nordic countries."

Reviewer: 3

Dr. Ludoviko Zirimenya, MRC/UVRI and LSHTM Uganda Research Unit Comments to the Author:

Generally, this is an exciting area of focus and the researchers have invested a lot of time, resources, and energy to undertake the study.

Response 14: We thank the reviewer for the positive view.

However, this manuscript is not well structured so it is difficult to understand.

The abstract is not well structured, the author is referred to the journal's author submission guidelines that clearly show how it should be structured.

Response 15: We recognize that the structure of this manuscript is indeed different from the IMRAD basic structure and does not follow the STROBE guidelines. As a combined reply to many of the following points raised by the reviewer, we kindly appoint the reviewer's attention to the type of manuscript that we present, which is a Cohort profile article, it does not fall within the Original research category. The content and structure of both main text and abstract is according to the guidelines for Cohort profiles provided by BMJ Open. The type of manuscript is stated in the title, we have further added the aim of the cohort profile to the introduction to make it clearer what the reader can expect from this profile, please see Response 13 and Response 24.

The researchers should use the IMRAD basic structure of scientific papers. The introduction section is well written but unfortunately the methods, results, and discussion sections, need to be improved. For example, the section on cohort description covers both description of the methods and results, these should be separate. When I read the abstract, I thought that the manuscript describes a protocol of planned work, but on further reading, they share some findings one wonders how they are linked to non-specific effects of childhood vaccinations. Clarity is needed on what exactly this manuscript is intended to be.

Response 16: We thank the reviewer for specifying the areas that cause confusion. We have structured the section "cohort description" following the headlines for Cohort Profiles provided in BMJ Open's author guideline, which reads "Cohort description Describe the setting, locations and relevant dates, including periods of recruitment, exposure, follow-up and data collection. Give the eligibility criteria and how participants were recruited. Report numbers of individuals at each stage of the study, e.g. how many were approached, included in the study and have been retained. Reasons for non-participation should be reported. A flow diagram is recommended to illustrate this. Describe methods of data collection and follow-up, and any external data sources used. Give characteristics of study participants (e.g. demographic, clinical, social) and information on exposures and potential confounders. Indicate number of participants with missing data for each variable of interest. Detailed statistical plans should not be reported."

It is further correct that we share some findings. This is also in alignment with the required content from the BMJ guidelines " Findings to date Include a short explanation of the most notable results from the cohort so far, with references to relevant publications. This section should summarise rather than present results." this section has further been updated to focus on the more general lines of the progress of the project: please also see Response 5. Finally, we have added the aim of the cohort profile to the introduction to clarify what this manuscript is intended to be (please see Response 13 and Response 15)

Please be aware of the tense so that it does not read as if this is a protocol paper. E.g. We will analyse.... Is better phrased as we analyzed etc.

Response 17: Future tense has been applied, as we are indeed describing future studies that are yet to be undertaken.

The main aim is clearly mentioned but what are those specific research questions mentioned in line 53 that guided the analysis?

Response 18: We have updated the text so it now reads "the individual studies will be undertaken using the same methodology and statistical coding across countries. Furthermore, we will examine

the same research question in multiple studies using different analytical approaches to facilitate triangulation of the results”

As per the STROBE guidelines, the methods section can be better described to make the methods more understandable. Please include items such as possible sources of bias and how they were addressed, statistical methods used, etc. such aspects are missing and will help a reader understand the paper better.

Response 19: This cohort profile covers the entire NONSEnse project and cohort, with many different studies attached. Going into detail of the specific bias pertained to each individual study within the project, is beyond the scope of this paper but will be described in detail in each of the associated studies. Instead, we focus on some general thoughts on the possibilities of investigating non-specific effects of vaccines using this observational registry data, methodological considerations, and comparability and quality of the included data from the different data sources.

Furthermore, the structure of the manuscript should be written to align with the STROBE guidelines. E.g. the discussion of the results is missing to highlight how the results were interpreted and their generalizability etc. Please include a supplementary report of its checklist.

Response 20: The STROBE guideline is not directly applicable for Cohort profiles as described in Response 15, e.g. we are not presenting any study specific results that can be discussed for their generalizability, we are merely describing the content of the data obtained from the different registries across the Nordic countries.

Prior to the start, where the criteria set out of who to include and exclude from the study to avoid any bias? As well description of methods of follow-up should be shared.

Response 21: Inclusion, exclusion, and follow-up will be defined independently for each of the different studies within the projects and will thus be described in connection with these.

Were any ethical approvals obtained for the study in all the countries from regulatory and ethical committees? How was access to data granted and confidentiality maintained?

Response 22: We thank the reviewer for highlighting this essential part. We have now added a section for ethical approvals following a section with information on storing and analysing the data (see Response 11). The ethics section reads “Ethical approval is not required for registry-based studies in Denmark and Finland. However, the study was approved by the Danish Data Protection Agency and by the Institutional Review Board of the Finnish Institute for Health and Welfare. In Norway, study approval was obtained from the Regional Ethics Committee, South-East. In Sweden, study approval was obtained from the Regional Ethical Review Board, Stockholm, Sweden.”

Reviewer: 4

Angela Pinot de Moira , University of Copenhagen Comments to the Author:

Cohort Profile: Childhood morbidity and potential nonspecific effects of the childhood vaccination programmes in the Nordic countries (NONSEnse): Register-based cohort of children born 1990-2017/2018.

The paper describes the creation of a Nordic register-based cohort to investigate the non-specific effects of vaccines on the overall health of children. To create the cohort, health and sociodemographic register data from 9,072,420 children aged 0-17 years born between 1990 and 2018 and registered as living in Denmark, Finland, Norway or Sweden were harmonized according to a Common Data Model.

The cohort will be a useful resource for examining the effects of vaccines on children's health and the paper is well written. However, I have some recommendations for improvement before the paper is published.

Response 23: We thank the reviewer for the positive feedback, and useful comments provided.

Major Comments

1) Although the aims of the NONSense project are clearly described, it would be useful to also set out the aims of the paper. I assume these are to describe how the register-based cohort was created, the characteristics of the cohort and to give an overview of the data included.

Response 24: As described in Response 13, we have now added aim of the paper to the introduction.

2) The Lexis diagrams are very helpful in obtaining an overview of available data, but in addition to these, it would be useful if the authors could explain the rationale for their chosen start date (1990) of follow-up. Vaccination data are only available from 1995 in Norway, later in the other countries and in Sweden no vaccination data were available before 2013. Since children were only followed until 18 years, children born 1990-1995 in Sweden will have no vaccination data. Similarly, hospital contacts have only reached national coverage since 2008 in Norway, meaning that many children will be missing these data. I wonder if a later start date would have offered more consistency across the countries.

Response 25: We have now added "The individual registries included in this cohort were established in the respective countries at different time points. We have included the birth cohorts from 1990 in all countries to ensure that we have full information on follow-up from birth also for the children who will be included at older ages for e.g., the studies of HPV vaccination given to teenagers.

3) More information on the harmonization process, the Common Data Model and the selection of variables in the main section of the manuscript would be useful.

Response 26: Due to space considerations in this already lengthy manuscript, we have only added a few additional lines to this description under source and content of data "The included data reflects necessary information to identify vaccination status of the child, relevant vaccine non-targeted outcomes, potential confounding factors, and information to be included as negative control outcomes". If the editor and reviewer wish further elaboration, we are happy to provide this. Also, for comparability across countries, it would be useful if the tables included in the "NONSense Common Data Model" document could be combined for the four countries (i.e. one table for prescriptions, one table for hospital contacts etc.).

Response 27: We thank the reviewer for this suggestion. We have updated the appendix to include all countries in the same table for comparison of source of data in each country.

Minor comments

1) In the abstract, under "Findings to date", the authors state: "we have identified indicators measuring similar levels of infectious disease morbidity across these settings. We have also conducted an interrupted-time series analysis of the association between the second dose of measles-mumps-rubella (MMR) vaccination and the rate of infectious disease hospitalisations due to infections not targeted by the MMR vaccine", however, these findings are presented in other papers and only discussed in the current paper.

Response 28: The section and heading "findings to date" is per the guidelines provided by BMJ Open. We have updated this paragraph in this revised version, to focus more on the overall progress of the projects (also see Response 5). If the Editor has further comments on how to include the findings to date differently in this manuscript, we will happily update it further.

2) I would rephrase the aim on lines 45-47 of the introduction.

Response 29: We have updated the description of the aim to “The main aim of NONSEnse is to evaluate if childhood vaccinations influence other health outcomes than those targeted by the vaccine in the Nordic countries”

3) Page 8, line 49: why the different follow-up end date in the four countries?

Response 30: The different end of follow-up is due to prolonged waiting times for getting the necessary approvals and obtaining data in each country, thus in some countries we could get data until 2018, reflecting when the approvals were obtained. We have added the following text to the manuscript “End of follow-up in each country reflects when the data application process was final.”

4) Page 10, line 50: Finland to (rather than Finlandto).

Response 31: Thanks, it has been corrected.

5) Page 18, line 36: although given in Table 4, it would be useful to also clarify the definition of low birthweight in the text.

Response 32: Thanks, we have added the description in main text as instructed.

6) Could the large amount of missing data on maternal smoking in Norway also partly explain the lower prevalence of maternal smoking in this country?

Response 33: We share the belief that that smoking mothers may be more likely to have missing information on smoking than non-smoking mothers, which could explain the low prevalence of maternal smoking. We have added the following text to the manuscript: “However, the greater proportion with missing information on maternal smoking in Norway could partly explain the lower proportion with registered maternal smoking during pregnancy, if missing information is more prevalent among smoking mothers.”

7) Page 25: am I right in understanding that these data are not open to other researchers to utilise, even for researchers based within the Nordic countries?

Response 34: Correct, it is not legal to share this data with other researchers that are not part of the project group, even if they do fulfil the requirements for using this data. Meanwhile, researchers that fulfil the requirements for using registry data can apply for the same data and prepare it using the concepts presented in the common data model. We have updated the data sharing statement, to elaborate on this point Please also see Response 3, Response 4, and Response 12

Reviewer: 5

Dr. Aditi Dey, National Centre for Immunisation Research and Surveillance of Vaccine Preventable Diseases

Comments to the Author:

Thanks for giving me the opportunity to review the manuscript titled, “Cohort Profile: Childhood morbidity and potential non-specific effects of the childhood vaccination programmes in the Nordic countries (NONSEnse): Register-based cohort of children born 1990-2017/2018”. I commend the authors for undertaking this important piece of research.

Response 35: We thank the reviewer for the positive feedback to the manuscript

However, the manuscript appears very long and had gaps in information on the non-specific effects of the childhood vaccination programmes.

Response 36: This manuscript is a cohort profile and not a single study. We agree that it is quite lengthy, which reflects that we are describing a comprehensive cohort including detailed information on the data from multiple registries from 4 different countries. We do not believe the manuscript can be shortened without losing content. It is the aim of the project to investigate the non-specific effects of the vaccination programmes – the studies to investigate non-specific effects conducted so far only pertain to an interrupted time series analysis of MMR vaccination. Future studies will investigate non-specific effects of vaccines in more detail as described.

I would have preferred to see what were the outcome measures (non-specific effects) that the researchers were investigating. How were these non-specific defined/ascertained/measured?

Response 37: We thank the reviewer for the comment. The measures of non-specific effects are included at the end of the introduction where we write “ The main associations we will examine are

associations between childhood vaccinations and 1) infectious disease hospitalisations, 2) antibiotic use, and 3) atopic diseases (asthma, atopic dermatitis, allergic rhinoconjunctivitis).“ We further describe how the specifics of these measures will be determined following an investigation of these outcomes in all countries: “The first step has been to examine and compare infectious disease and atopic morbidity among children in the respective countries over time and by age and sex, to inform choice of design and outcome definitions in the subsequent studies of non-specific effects of vaccines.”

How long was the follow-up period for each of these non-specific effects to develop and or resolve?
How were confounders accounted for?

Response 38: The specifics for definition of outcome, confounders, and follow-up, will vary from study to study depending on the research question we seek to answer. Thus, these will be defined individually in the specific studies within the project.

The authors mention in the abstract about undertaking a time-series analysis of the association between the second dose of measles-mumps-rubella (MMR) vaccination and the rate of infectious disease hospitalisations due to infections not targeted by the MMR vaccine. Again, I was unable to find these results in the main body of the manuscript.

Response 39: We thank the reviewer for this comment, which has also been raised by more of the other reviewers (see Response 5, Response 16, and Response 28). We have updated the content described in this section, and hope that it is now clearer what is described.

Ethics approval processes were not mentioned in the manuscript though the authors mentioned that due to national data protection legislation, country-specific data were stored and analysed in the respective countries.

Response 40: We thank the reviewer for highlighting this essential part of the manuscript, which has also been noted by other reviewers. We have added a section on ethics. Please see Response 11 and Response 22 for further details.

Reviewer: 1

Competing interests of Reviewer: None.

Reviewer: 2

Competing interests of Reviewer: No competing interest

Reviewer: 3

Competing interests of Reviewer: None

Reviewer: 4

Competing interests of Reviewer: None

Reviewer: 5

Competing interests of Reviewer: None

VERSION 2 – REVIEW

REVIEWER	Abtahi , Shahab Universiteit Utrecht, Division of Pharmacoepidemiology and Clinical Pharmacology, Utrecht Institute for Pharmaceutical Sciences
REVIEW RETURNED	11-Jan-2023

GENERAL COMMENTS	Dear Editor, I think the revised version of this paper addressed all my previous concerns or questions and can be considered by you for publication. No further comments from me. Best wishes, Shahab Abtahi, MD PhD Utrecht University
---

REVIEWER	Pinot de Moira , Angela University of Copenhagen
REVIEW RETURNED	21-Dec-2022

GENERAL COMMENTS	The authors have clearly considered the suggested amendments and I am largely happy with the specific responses to my comments. Regarding the "Findings to date" section in the abstract: perhaps a more informative solution would be to briefly describe the characteristics of the cohort (median follow-up time, vaccination coverage etc.), thus summarising the "Cohort Description" from the main text of the manuscript? In addition, it would be useful to include a short sentence on the aims of the paper in the abstract (under purpose), similar to that now included in the introduction.
--

REVIEWER	Dey, Aditi National Centre for Immunisation Research and Surveillance of Vaccine Preventable Diseases
REVIEW RETURNED	20-Dec-2022

GENERAL COMMENTS	Thanks for revising the manuscripts based on feedback of reviewers. My previous queries, "How long was the follow-up period for each of these non-specific effects to develop and or resolve? How were confounders accounted for?" were not addressed in the manuscript though the authors have responded to it in their reply. It would be good to include this in the main body of the manuscript. Greater clarity would be achieved if a range of follow-up period, potential confounders and outcomes were provided in the manuscript.
--